# Treatment Monitoring of a Patient with Synchronous Metastatic Angiosarcoma and Breast Cancer Using ctDNA

**DOI:** 10.3390/ijms25074023

**Published:** 2024-04-04

**Authors:** Christoffer Vannas, Mandy Escobar, Tobias Österlund, Daniel Andersson, Pia Mouhanna, Amanda Soomägi, Claes Molin, David Wennergren, Henrik Fagman, Anders Ståhlberg

**Affiliations:** 1Sahlgrenska Center for Cancer Research, Department of Laboratory Medicine, Institute of Biomedicine, Sahlgrenska Academy, University of Gothenburg, 40530 Gothenburg, Sweden; mandy.escobar@gu.se (M.E.); tobias.osterlund@gu.se (T.Ö.); daniel.andersson.3@gu.se (D.A.); pia.mouhanna@gu.se (P.M.); amanda.soomagi@gmail.com (A.S.); henrik.fagman@gu.se (H.F.); 2Department of Oncology, Sahlgrenska University Hospital, Region Västra Götaland, 41345 Gothenburg, Sweden; claes.molin@vgregion.se; 3Department of Clinical Genetics and Genomics, Sahlgrenska University Hospital, Region Västra Götaland, 41345 Gothenburg, Sweden; 4Wallenberg Centre for Molecular and Translational Medicine, University of Gothenburg, 40530 Gothenburg, Sweden; 5Department of Oncology, Ryhov County Hospital, 55185 Jönköping, Sweden; 6Department of Orthopaedics, Institute of Clinical Sciences, Sahlgrenska Academy, University of Gothenburg, 40530 Gothenburg, Sweden; david.wennergren@vgregion.se; 7Department of Clinical Pathology, Sahlgrenska University Hospital, Region Västra Götaland, 41345 Gothenburg, Sweden

**Keywords:** circulating tumor DNA, liquid biopsy, sarcoma, angiosarcoma, treatment monitoring, personalized medicine

## Abstract

Angiosarcoma is a rare and aggressive type of soft-tissue sarcoma with high propensity to metastasize. For patients with metastatic angiosarcoma, prognosis is dismal and treatment options are limited. To improve the outcomes, identifying patients with poor treatment response at an earlier stage is imperative, enabling alternative therapy. Consequently, there is a need for improved methods and biomarkers for treatment monitoring. Quantification of circulating tumor-DNA (ctDNA) is a promising approach for patient-specific monitoring of treatment response. In this case report, we demonstrate that quantification of ctDNA using SiMSen-Seq was successfully utilized to monitor a patient with metastatic angiosarcoma. By quantifying ctDNA levels using 25 patient-specific mutations in blood plasma throughout surgery and palliative chemotherapy, we predicted the outcome and monitored the clinical response to treatment. This was accomplished despite the additional complexity of the patient having a synchronous breast cancer. The levels of ctDNA showed a superior correlation to the clinical outcome compared with the radiological evaluations. Our data propose a promising approach for personalized biomarker analysis to monitor treatment in angiosarcomas, with potential applicability to other cancers and for patients with synchronous malignancies.

## 1. Introduction

Angiosarcoma is an aggressive soft-tissue sarcoma of vascular origin that constitutes approximately 2% of all soft-tissue sarcomas [1]. It exhibits a high propensity for metastasis, commonly to the lungs and, to a lesser extent, to other soft-tissue compartments, viscera and bones [2]. The prognosis is consequently dismal with a 30–40% 5-year survival rate [3,4]. Known risk factors to develop angiosarcomas include chronic lymphedema, previous radiation therapy and long-term exposure to certain toxins, such as vinyl chloride and arsenic [5,6,7,8]. Hereditary conditions including Neurofibromatosis type 1, Maffucci syndrome, Klippel–Trénaunay syndrome and BRCA1/2-mutations are also associated with an increased risk [9,10].

Angiosarcoma can be subdivided into cutaneous angiosarcoma and angiosarcoma of soft tissues [2]. Cutaneous angiosarcoma typically manifests as expanding purple skin papules, while angiosarcoma of soft tissues presents as expanding masses at the respective site of origin. Histologically, angiosarcoma displays multinodular hemorrhagic masses with atypical epithelioid or spindle cells, featuring high mitotic activity. The neoplastic cells form irregular, primitive vascular channels, infiltrating the surrounding tissues. Vascular markers, such as CD31, CD34 and ERG, are often highly expressed and may be used in diagnosis [2]. Angiosarcomas harbor multiple genomic alterations, with recurrent mutations in *TP53*, *CDKN2A*, *MYC* and genes involved in MAPK signaling [11].

Localized angiosarcoma is treated with surgical resection, aiming for a wide resection margin. The benefit of adjuvant therapy is uncertain, but many patients receive chemotherapy, sometimes combined with radiation therapy, based on national guidelines [3]. Metastatic angiosarcoma is primarily treated with chemotherapy, where anthracycline- or taxane-based regimens are most commonly used [12]. Different inhibitors of tyrosine kinase receptors or ligands have been tested in clinical trials with modest effects [13,14,15], while immune checkpoint inhibitors using PD-1 and CTLA-4 inhibitors have shown more promising effects [16].

Monitoring of angiosarcoma relies primarily on clinical and radiological evaluations without any well-established blood-based marker in use for treatment efficacy monitoring. Lactate dehydrogenase (LDH) serves as a non-specific cancer marker that is often elevated with increased tumor burden [17]. LDH quantification for treatment monitoring in sarcomas is sporadically conducted at certain centers according to local clinical guidelines. Additionally, quantification of albumin levels in blood plasma is performed as a general indicator of nutritional and disease status, typically showing a decline in cancer patients with progressive disease [18]. However, both LDH and albumin possess limited sensitivity and specificity, rendering them unsuitable for routine treatment monitoring of sarcomas.

The detection and quantification of circulating tumor DNA (ctDNA) in blood plasma present a promising method for treatment monitoring in patients with solid tumors. Nevertheless, few studies focused on angiosarcoma have been published to our knowledge. In a study by Tanaka et al., ctDNA was successfully identified in four out of five patients with metastatic angiosarcoma using a generic ctDNA panel targeting recurrent cancer mutations [19]. In one patient, longitudinal quantification of three patient-specific mutations using digital PCR displayed a correlation between ctDNA levels and clinical outcome. The limitations with this approach are that few mutations can be detected with generic panels and that digital PCR can only assess a few patient-specific mutations [20]. An alternative approach is to identify patient-specific mutations in tumor tissue by whole-exome sequencing at diagnosis. Based on these mutations, personalized ctDNA panels are then designed to monitor patients over time [21]. SiMSen-Seq is an ultrasensitive sequencing technique that is suitable for personalized ctDNA analysis, enabling the detection of less than 0.1% variant allele frequencies [22]. Previous studies have demonstrated the effectiveness of SiMSen-Seq in monitoring treatment responses in cancer patients [23,24,25,26].

Illustrated by this case, we demonstrate how personalized ctDNA analysis can be utilized to monitor treatment response and predict the outcome in a patient with synchronous metastatic angiosarcoma and metastatic breast cancer.

## 2. Case Presentation

In July 2019, an 83-year-old woman with a history of metastatic breast cancer, displaying long-term stable disease on endocrine therapy, underwent assessment at Sahlgrenska University hospital. She had a progressively enlarging cutaneous mass in the right axilla/humerus that had developed over six weeks (Figure 1A). Initially diagnosed as erysipelas, the persistence of the mass despite multiple antibiotic treatments prompted a biopsy, revealing a high-grade angiosarcoma.

The patient underwent surgery for the tumor, involving extensive tumor excision and a split-thickness skin graft. She recovered swiftly after surgery and showed good performance status at the first follow-up after surgery. Pathological analysis of the resection specimen revealed a non-radically excised high-grade angiosarcoma with a maximum diameter of 155 mm (Figure 1B). The patient was admitted for adjuvant chemotherapy, but before any treatment was initiated, the patient experienced a local tumor recurrence in the skin graft and multiple cutaneous satellite metastases proximally and distally to the resection margin (Figure 1C). Consequently, an interscapular–thoracic amputation of the right arm was performed. The patient was again admitted to adjuvant chemotherapy, but before the initiation of the treatment, she was hospitalized for pneumonia. A lung computed tomography (CT) scan was performed that revealed multiple lung metastases (Figure 1D, day 88). Following recovery from the pneumonia, palliative chemotherapy with paclitaxel was initiated, which was tolerated with no severe side effects. After five cycles of chemotherapy, a surprising complete regression of the lung metastases was observed on a lung CT scan (Figure 1D, day 204).

Following the sixth cycle of chemotherapy, the patient required a treatment respite due to fatigue. A three-month treatment cessation followed by radiological evaluation was planned. However, one month after the last cycle of chemotherapy, the patient was hospitalized due to cauda equina syndrome. A whole-spine magnetic resonance imaging displayed new metastases in cervical, lumbar and sacral vertebrae with a spinal canal obliteration in the sacrum. A lung CT scan was performed, which indicated bilateral pleural effusion and multiple new lung metastases (Figure 1D, day 231). A biopsy from a metastatic lesion in a thoracic vertebra initially indicated a relapse of breast cancer but was later confirmed as an angiosarcoma relapse after immunohistochemical staining (Figure 1E). Palliative radiotherapy to the L5-S1 vertebrae was administered, but the patient succumbed to the disease only three weeks later. 

Before the second surgery, the patient was included in the SARKOMTEST-study, a study exploring the potential value of blood-based markers, especially ctDNA, for monitoring treatment efficacy in patients diagnosed with sarcoma. Blood plasma collected before and after the second surgery, at each chemotherapy cycle and during follow-up was stored for retrospective ctDNA analysis (Figure 1F). Whole-exome sequencing of the primary tumor from the first surgery was performed and gene-set enrichment analysis of the 500 mutations with the highest variant allele frequencies displayed genes involved in metastasis, epithelial–mesenchymal transition, UV radiation response and neurogenesis (Appendix A). Twenty-five patient-specific mutations with high allele frequencies in the tumor tissue were selected for ctDNA monitoring using SiMSen-Seq (Appendix A). The selected genes belonged to several different gene-set enrichments (Appendix A).

All patient-specific mutations were detectable in blood plasma before the second surgery and their levels showed no post-surgical decrease. On the contrary, the levels were 3.8-fold higher after surgery, suggesting an alternative source of ctDNA, possibly pre-existing metastases (Figure 2A, Appendix A). The ctDNA levels had increased further at initiation of palliative chemotherapy and remained detectable during all chemotherapy cycles, even when the radiological evaluation indicated a complete response (Figure 2A, Appendix A). Before the initiation of radiation therapy, the ctDNA levels increased even further, consistent with radiological evidence of tumor progression. Interestingly, one day after the radiation therapy, the ctDNA levels displayed a 2.6-fold increase. (Figure 2A, Appendix A). We then analyzed the number of ctDNA mutations that increased or decreased over time and found a predominance of escalating dynamics at all time points, except between chemotherapy cycles 2–4. At chemotherapy cycle 3, the levels of all mutations temporarily decreased (Figure 2B), with an average 20 % decrease compared to chemotherapy cycle 1 (Appendix A).

The ctDNA level is often reported as variant allele frequency, which is the ratio between mutated ctDNA molecules and total cell-free DNA molecules (Figure 2C, Appendix A). This estimate compensates for technical errors when handling different samples. However, the levels of wildtype molecules in plasma are not constant and may change due to several physiological factors, such as infection and exercise, which will confound the frequency [27,28]. Here, the level of total cell-free DNA changed over time, with a 9.2-fold increase from the first and lowest to the last and highest time point. A 3-fold increase in cell-free DNA was observed both the day after surgery and the day after initiation of radiation therapy (Figure 2D). The alternative strategy is to report ctDNA levels as the number of ctDNA molecules per ml of plasma. This approach will not be affected by variations in the number of wildtype molecules. For this patient, ctDNA levels reported as mutant molecules per ml of plasma correlated better to the clinical outcome than variant allele frequency. Interestingly, the total number of white blood cells for this patient fluctuated 3.6-fold from the lowest to the highest value over time, which may have contributed to the pool of wildtype molecules and impacted the variant allele frequency data (Figure 2E).

The clinical routine markers LDH and albumin were also assessed over time (Figure 2F). LDH levels were assessed at two time points, prior to the second surgery and after the completion of radiation therapy. At both time points, the LDH levels were elevated; however, no dynamic indicative of tumor progression was observed. Albumin levels were assessed during the entire treatment period, revealing pathologically low levels at all time points. However, rapid decreases in albumin levels were observed following surgery and a couple of days before the patient’s death.

Finally, we investigated whether the mutations with the highest variant allele frequency in the tumor tissue also exhibited the most ctDNA molecules in plasma. Minimal correlation was observed between mutations with high frequency in tissue and the numbers of ctDNA molecules. In plasma, we detected high heterogeneity, but the mutations with the highest counts per ml of plasma prior to the second surgery, *HSPG2*, *SLC22A15* and *LIMCH1*, remained the highest expressed over time (Figure 3).

## 3. Discussion

This case highlights an interesting clinical course of a patient with metastatic angiosarcoma, demonstrating an initially remarkable radiological response to palliative chemotherapy. Regrettably, a swift relapse occurred following the discontinuation of treatment. The rapid disease progression after tumor recurrence aligns with the characteristic behavior of highly malignant tumors, such as angiosarcomas, underscoring the need for improved tools for early detection of disease progression and more effective therapeutic options.

Angiosarcomas often develop secondary to radiation therapy [5]. This patient had a history of radiation therapy administered to the right chest wall and axilla following a breast cancer, albeit slightly proximal to the origin of the angiosarcoma. Whether this was a radiation-induced angiosarcoma or not remains elusive. However, gene-set enrichment analysis of the mutated genes displayed UV-irradiated targets among the enriched gene sets, which aligns with the characteristics of a radiation-induced malignancy. It is possible that low doses of radiation impacted parts of the upper arm, either through scattered radiation or potential mispositioning of the arm during radiotherapy sessions.

Moreover, this case demonstrates the utility of treatment monitoring through quantification of ctDNA in blood plasma and its advantages to radiological evaluations and other blood-based markers like LDH and albumin. Despite extensive surgery, the ctDNA levels did not diminish, suggesting the presence of a residual subclinical tumor, contributing to the ctDNA pool, which was later radiologically confirmed. It is noteworthy that the ctDNA levels were in fact higher the day after surgery. This could be attributed to an increased secretion of ctDNA from a non-radically excised tumor lesion or by tumor cell death as a result of tissue ischemia during surgery. Decreased ctDNA clearance resulting from a temporal decline in renal function linked with hypovolemia after surgery may also have had an impact. Notably, the level of total cell-free DNA increased after surgery. At this time point, ctDNA only constituted a smaller part of the total cell-free DNA, indicating a major contribution of cell-free DNA from non-tumor cells. On the contrary, the rapid increase in cell-free DNA the day after radiation therapy was partly driven by an increase in ctDNA, supported by the high variant allele frequency of ctDNA.

Throughout palliative chemotherapy, ctDNA levels remained detectable, despite a radiologically confirmed complete tumor response. We cannot determine if the released ctDNA was due to tumor progression or the treatment. However, the swift relapse following the cessation of chemotherapy implies the existence of remaining metastases that rapidly progressed once the chemotherapy was discontinued. It is possible that these metastases were too small to be detected with a CT scan or located in an area that was not captured by the radiological evaluation. Employing alternative imaging modalities, such as magnetic resonance imaging or positron-emission tomography, for radiological monitoring may have facilitated disease tracking, enabling the observation of progression at an earlier stage. However, these modalities are time consuming, expensive and may necessitate extensive preparations from the patient. In contrast, ctDNA analysis offers a cost-effective, easily performed and minimally discomforting alternative for treatment efficacy monitoring.

At the time of radiological recurrence, the high ctDNA level strongly indicated an angiosarcoma relapse rather than a recurrence of breast cancer since patient-specific mutations from the angiosarcoma were detected. Although the impact of ctDNA dynamics on treatment decisions and outcome in this case remains uncertain, it is possible that the additional insight derived from the ctDNA data could have influenced the decision to perform a tumor biopsy of the vertebra. Biopsies are associated with risk and distress, particularly for patients with low performance status, severe symptoms or undergoing anticoagulant treatment. Opting not to conduct a tumor biopsy for this patient would likely have been correlated with an enhanced quality of life and an earlier referral to palliative care.

The potential for treatment monitoring using our approach opens up new avenues for clinical decision making. Implementing this method in a real-time setting enables the evaluation of treatment responses after short-term interventions, eliminating the need to wait for a standard radiological follow-up, typically conducted three months into treatment. A real-time evaluation could prove instrumental in tailoring treatment strategies for patients, especially in the context of clinical trials focused on rare tumor entities like angiosarcomas, with limited treatment options. It is imperative to emphasize that the sensitivity and specificity of ctDNA detection using SiMSen-Seq requires further evaluation, particularly in patients with lower metastatic tumor burden, localized disease, or slow-growing tumor entities. Additionally, the optimal number of mutations for ctDNA monitoring using SiMSen-Seq requires additional studies. In this case, we used 25 patient-specific mutations to enhance the sensitivity to detect ctDNA, observe increases and decreases in ctDNA levels, and minimize the risk of confounding data due to tumor heterogeneity. Interestingly, minor correlations were observed between the mutations with a high frequency in the tumor tissue compared with liquid biopsy, underscoring the difficulty of choosing an optimal mutation candidate or determining the appropriate number of mutations for ctDNA monitoring. To gain deeper insight into the tumor heterogeneity, it would have been interesting to sequence the local tumor recurrence and the biopsy from the bone metastasis as well as the original breast cancer. This may have provided information about clonal expansions across different anatomical sites and tumor entities. Based on our data, we cannot exclude the possibility that the patient had a simultaneous progression of the breast cancer, which was not histologically confirmed. Detailed ctDNA monitoring using patient-specific mutations from the different tumor locations may have been even more informative.

In conclusion, ctDNA quantification using a patient-specific mutation panel displayed potential for improved treatment monitoring of cancer patients. In this case, the additional data derived from ctDNA quantification provided early insights into the surgical outcome, improved treatment monitoring compared to radiological evaluations during palliative chemotherapy and facilitated the monitoring of one specific tumor entity in a patient with two synchronous cancers. Further research is warranted to refine and optimize this approach for use in clinical routine applications.

## 4. Materials and Methods

### 4.1. Blood Sample Collection

The patient participated in the SARKOMTEST-study and signed an informed consent form. This study was approved by the Regional Ethical Review Board in Gothenburg (485-16, T-795-15, T525-18 and 2021-04895). Plasma was isolated within 2 h from blood collected in EDTA tubes, with centrifugation at 2000× *g* for 10 min. Plasma and buffy coat were stored at −80 °C. Blood samples for quantification of WBC, LDH and albumin were collected and analyzed in clinical routine. All radiological assessments were performed according to clinical routine.

### 4.2. Whole-Exome Sequencing

Tumor DNA was extracted from formalin-fixed paraffin-embedded material using GeneRead FFPE DNA kit (Qiagen, Hilden, Germany) and DNA was extracted from buffy coat using a QIAamp DNA Blood Mini Kit (Qiagen). Sequencing was performed by the SNP&SEQ Technology Platform in Uppsala.

Raw *fastq*-files were used as input to the Sarek pipeline [29] in somatic variant calling mode where the tumor sample was matched to the corresponding buffy coat DNA sample. The Sarek pipeline aligned the reads to the hg19 reference genome and identified single-nucleotide variants, insertions and deletions using Mutect2 [30], Strelka2 [31] and Freebayes [32] for somatic variant calling. High-confidence exonic variants that were identified by all three callers were used for downstream analysis.

### 4.3. Digital ctDNA Sequencing

Circulating cell-free DNA was extracted from blood plasma with the QIAsymphony system using QIAsymphony DSP Circulating DNA Kit (Qiagen), according to the manufacturer’s instructions. The concentration of cell-free DNA was quantified with Qubit 3.0 Fluorometer (Thermo Fisher Scientific, Waltham, MA, USA). SiMSen-Seq was used to analyze circulating cell-free DNA as described [33] with some adjustments. First, we designed a SiMSen-Seq panel targeting 25 mutations with high variant allele frequency, based on the exome sequencing data. Then, libraries for sequencing were generated using SiMSen-Seq. SiMSen-Seq consists of two sequential PCR steps. In the first step, each target molecule is barcoded with a unique molecular identifier. In the second step, targeted and barcoded molecules are amplified with indexed Illumina adapters. Library quality was assessed with a 5200 Fragment Analyzer System with the HS NGS Fragment Kit (both Agilent technologies, Santa Clara, CA, USA) according to the manufacturer’s instructions. The libraries were then pooled and purified with the Pippin Prep DNA Size Selection System using the 2% Agarose kit (both Sage Science, Beverly, MA, USA). Sequencing was performed on the MiniSeq Sequencing System using 20% PhiX control v3 and High Output Reagent Kit (both Illumina, San Diego, CA, USA) in single-end 150 bp mode. Sequencing data was processed using the UMIErrorCorrect pipeline [34]. A minimum of three reads per unique molecular identifier was applied to form consensus reads. Data are reported as both mutated molecules per milliliter of plasma and variant allele frequencies.

### 4.4. Data Analysis

Gene-set enrichment analysis was performed using GenePattern 2.0 on the 500 mutations with the highest variant allele frequency from the whole-exome sequencing of the tumor tissue [35]. Figures were created using GraphPad prism 10.1.2 (GraphPad, San Diego, CA, USA). The radiological response to treatment was evaluated according to RECIST 1.1. All ctDNA values are shown in Appendix A.

## Figures and Tables

**Figure 1 ijms-25-04023-f001:**
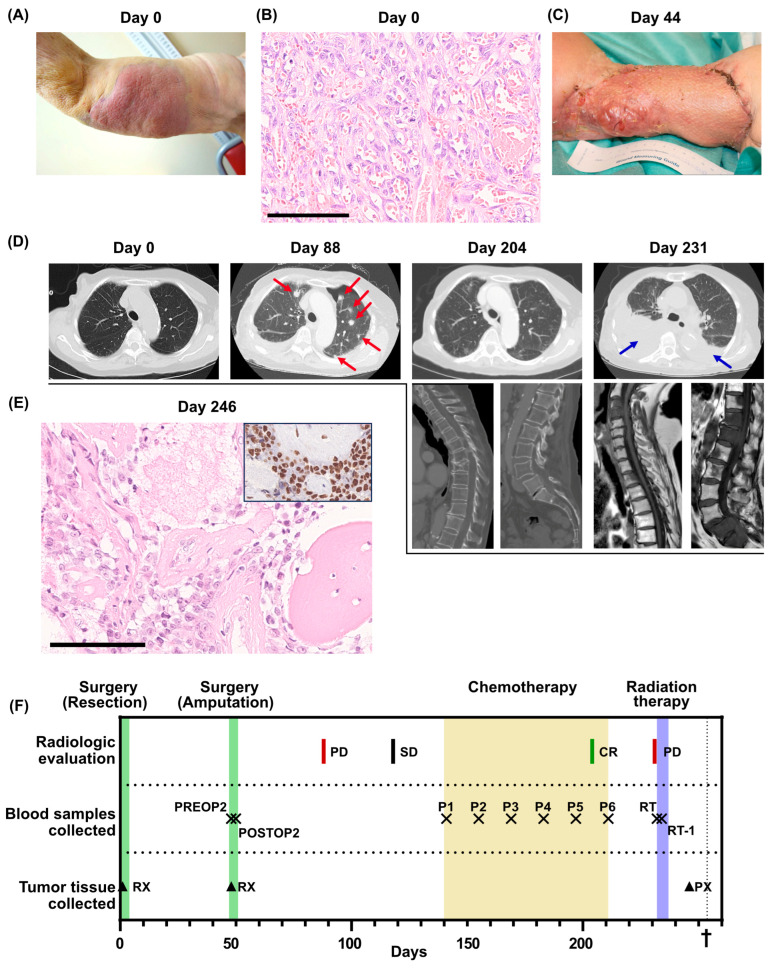
Clinical overview. (**A**) Photo of the primary tumor before resection (day 0). (**B**) Histological section of the primary tumor. High-grade angiosarcoma with primitive vasoformative pattern. Mitotic figures are abundant. Scale bar 100 µm. (**C**) Photo of the arm after the first recurrence (day 44). (**D**) Radiological evaluations. For day 204 and day 231, multiple examinations were performed. Metastases are indicated by red arrows. Pleural effusion is indicated by blue arrows. (**E**) Bone metastasis of high-grade angiosarcoma. Insert shows ERG immunohistochemistry of the same metastasis. Scale bar 100 µm. (**F**) Overview of the clinical course, displaying treatments, radiological evaluations, when blood samples were collected for the SARKOMTEST-study and when tumor tissue was collected. PD, progressive disease; SD, stable disease; CR, complete remission; PREOP2, blood sample collected the day before the second surgery; POSTOP2, blood sample collected the day after the second surgery; P1–P6, blood samples collected before each cycle of palliative chemotherapy; RT, blood sample collected the day before initiation of radiation therapy; RT-1, blood sample collected the day after initiation of radiation therapy; RX, resection; PX, tumor biopsy. The dagger (†) indicates when the patient deceased.

**Figure 2 ijms-25-04023-f002:**
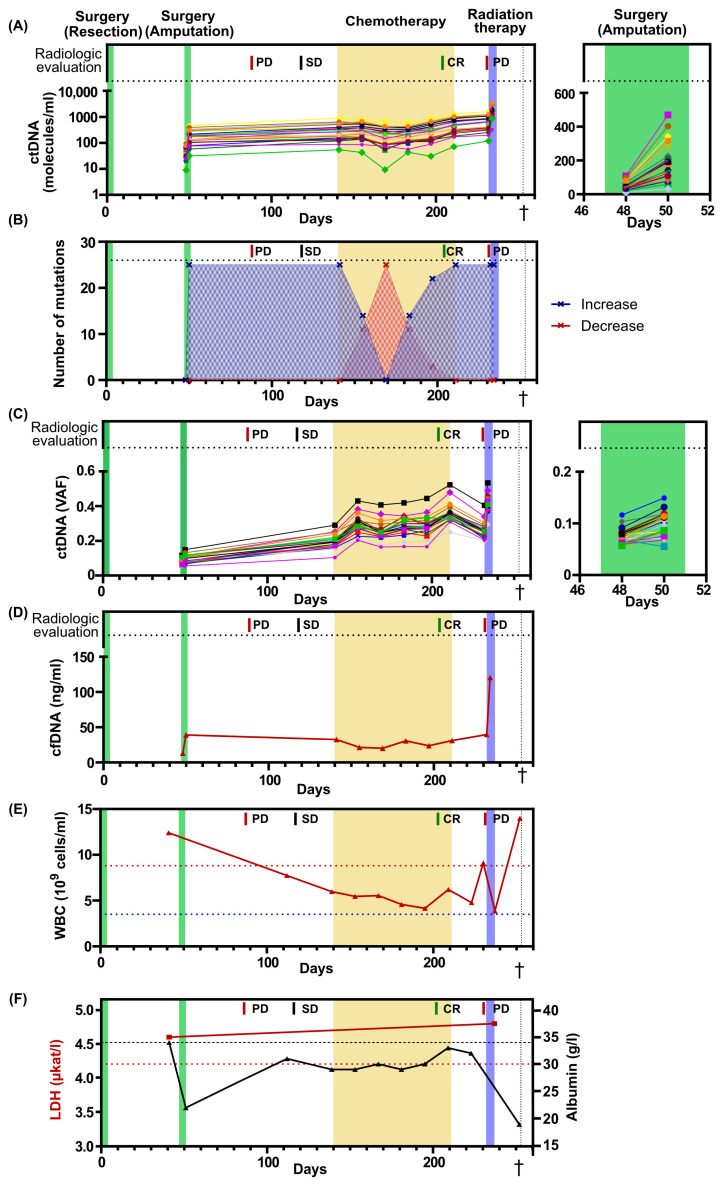
Blood-based markers over time. (**A**) The levels of ctDNA expressed as number of molecules per ml of plasma. In total, 25 different mutations were assessed. Treatments and radiological evaluations displayed at the top. On the right, a magnified view of the second surgery, spanning from day 46 to day 52. (**B**) Number of mutations that increase or decrease their levels in plasma compared with the previous time point. (**C**) The levels of ctDNA expressed as variant allele frequency (VAF). On the right, a magnified view of the second surgery, spanning from day 46 to day 52. The colors of each mutation in Figure 2A,C are identical. Each line corresponds to a specific gene. (**D**) Level of cell-free DNA (cfDNA) after plasma extraction. (**E**) The numbers of white blood cells (WBC) over time. Dotted blue line indicates lower limit of normal, while dotted red line indicates upper limit of normal. (**F**) The levels of LDH and albumin over time. Dotted red line indicates upper limit for normal LDH levels, while dotted black line indicates lower limit for normal albumin levels. PD, progressive disease; SD, Stable disease; CR, complete remission. The dagger (†) indicates when the patient deceased.

**Figure 3 ijms-25-04023-f003:**
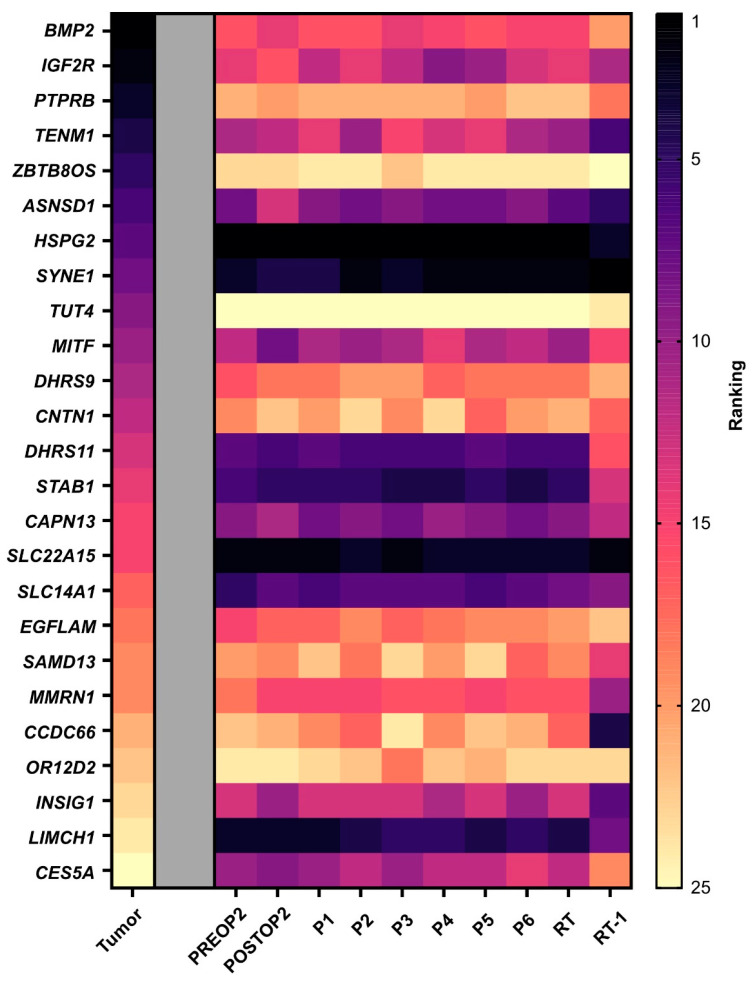
Tumor heterogeneity. A heat map displaying ctDNA heterogeneity in plasma. The mutations are ordered based on variant allele frequency in the tumor tissue, with the highest frequency ranked as number one. The mutations in the plasma samples are then ranked for each time point based on number of molecules per ml of plasma. Note that tumor tissue data are based on variant allele frequency since whole-exome sequencing only provides frequencies. PREOP2, blood sample collected the day before the second surgery; POSTOP2, blood sample collected the day after the second surgery; P1–P6, blood samples collected before each cycle of palliative chemotherapy; RT, blood sample collected the day before initiation of radiation therapy; RT-1, blood sample collected the day after initiation of radiation therapy.

## Data Availability

The data that support the findings of this study are available on request from the corresponding author, C.V. The GDPR legislation requires us to protect the identity of the participants, and the raw data cannot be publicly shared.

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
