# Peer review of "Treatment Monitoring of a Patient with Synchronous Metastatic Angiosarcoma and Breast Cancer Using ctDNA"

_ijms, 2024, doi:10.3390/ijms25074023_

Round 1

Reviewer 1 Report

Comments and Suggestions for Authors

Dear Authors,

1.  The main question addressed by the research is the utility of circulating tumor DNA (ctDNA) quantification for monitoring treatment efficacy and disease progression in a patient with metastatic angiosarcoma.

2.      Original and relevant aspects of the paper include:

-  The demonstration of ctDNA dynamics throughout various stages of treatment, including surgery, chemotherapy, and radiation therapy.

-   Comparison of ctDNA levels with traditional markers such as LDH and albumin.

-   Exploration of the correlation between tissue mutations and ctDNA levels.

-     Discussion on the potential implications of ctDNA monitoring for treatment decisions and patient outcomes. 

3.   The paper adds to the subject area by providing detailed insights into the clinical course of a patient with metastatic angiosarcoma and demonstrating the potential utility of ctDNA quantification for treatment monitoring. This adds to the existing literature by highlighting the advantages of ctDNA analysis over traditional radiological evaluations and other blood-based markers.

4.     Methodological improvements and further controls to consider:

-        Advise the authors to perform an whole-exome sequencing on tumour recurrence in the skin graft, where possible, in order to check for a more heterogeneous correlation between high-frequency mutations in the tissue and number of ctDNA molecules. In addition, checking for possible mutations between the primary tumour and the recurrence.

-     I would suggest that the authors elaborate on the functions and processes of the 25 genes chosen for the patient-specific panel in order to substantiate their choice in the tumour context of angiosarcoma. Adding a gene ontology panel to highlight the processes in which the genes are involved would increase the robustness of the study.

-    The authors might consider evaluating their mutational panel on already published databases on angiosarcoma to assess a possible tumor signature. 

5.    The conclusions drawn from the evidence and arguments presented in the article appear consistent. The main questions posed were addressed through specific experiments, including ctDNA quantification before and after various treatment interventions, comparison with traditional markers, and correlation analysis with tissue mutations.

6. The references seem appropriate, covering relevant literature on angiosarcoma, ctDNA analysis, and treatment monitoring strategies.

7.      Additional comments on tables, figures, and data quality:

-  The figures and tables are well-organized and effectively complement the text.

-   The inclusion of additional figures and tables enhances the understanding of the research results.

-      I would recommend adding a gene ontology panel for the 25 genes chosen for the specific patient panel, as mentioned above.

-      Regarding Figure 3, I would suggest adding an analysis of the tumor tissue from the recurrence and providing a more detailed explanation of the correlation between allele frequency in the tumor tissue and ctDNA trends in plasma.

Overall, the article provides valuable insights into the potential role of ctDNA quantification in monitoring treatment efficacy and disease progression in patients with metastatic angiosarcoma.

Author Response

Dear Authors,

  1.  The main question addressed by the research is the utility of circulating tumor DNA (ctDNA) quantification for monitoring treatment efficacy and disease progression in a patient with metastatic angiosarcoma.

Agreed

  1.      Original and relevant aspects of the paper include:

-  The demonstration of ctDNA dynamics throughout various stages of treatment, including surgery, chemotherapy, and radiation therapy.

-   Comparison of ctDNA levels with traditional markers such as LDH and albumin.

-   Exploration of the correlation between tissue mutations and ctDNA levels.

-     Discussion on the potential implications of ctDNA monitoring for treatment decisions and patient outcomes. 

Agreed

  1.  The paper adds to the subject area by providing detailed insights into the clinical course of a patient with metastatic angiosarcoma and demonstrating the potential utility of ctDNA quantification for treatment monitoring. This adds to the existing literature by highlighting the advantages of ctDNA analysis over traditional radiological evaluations and other blood-based markers.

Agreed

  1.     Methodological improvements and further controls to consider:

-        Advise the authors to perform an whole-exome sequencing on tumour recurrence in the skin graft, where possible, in order to check for a more heterogeneous correlation between high-frequency mutations in the tissue and number of ctDNA molecules. In addition, checking for possible mutations between the primary tumour and the recurrence.

Indeed, mutation analysis in the tumor recurrence sample is interesting and may provide clinical relevant information about clonal expansion related to treatment resistance. In this case study, it is not feasible to add this additional analysis. However, we have, updated the manuscript (Lines 287-294), pointing out that sequencing of the recurrence (and other tumor tissues) may provide additional information about tumor development and clonal expansions, as well as that ctDNA monitoring using mutations from all tumor locations may have been even more informative.

-     I would suggest that the authors elaborate on the functions and processes of the 25 genes chosen for the patient-specific panel in order to substantiate their choice in the tumour context of angiosarcoma. Adding a gene ontology panel to highlight the processes in which the genes are involved would increase the robustness of the study.

This is an insightful comment. We have included a new Supplementary Table 2B with information about the 25 genes and their link to different gene-set enrichments (Lines 136-137). When we performed gene enrichments using the 25 genes only, we received limited outputs since gene-set enrichment analysis is suitable for larger number of genes.

-    The authors might consider evaluating their mutational panel on already published databases on angiosarcoma to assess a possible tumor signature. 

This is good idea. However, it is out of scope for this case report. This will be relevant in future studies when analyzing additional angiosarcomas understanding the mutational landscape in more detail.

  1.   The conclusions drawn from the evidence and arguments presented in the article appear consistent. The main questions posed were addressed through specific experiments, including ctDNA quantification before and after various treatment interventions, comparison with traditional markers, and correlation analysis with tissue mutations.

Agreed.

  1. The references seem appropriate, covering relevant literature on angiosarcoma, ctDNA analysis, and treatment monitoring strategies.

Agreed

  1.      Additional comments on tables, figures, and data quality:

-  The figures and tables are well-organized and effectively complement the text.

-   The inclusion of additional figures and tables enhances the understanding of the research results.

-      I would recommend adding a gene ontology panel for the 25 genes chosen for the specific patient panel, as mentioned above.

-      Regarding Figure 3, I would suggest adding an analysis of the tumor tissue from the recurrence and providing a more detailed explanation of the correlation between allele frequency in the tumor tissue and ctDNA trends in plasma.

Overall, the article provides valuable insights into the potential role of ctDNA quantification in monitoring treatment efficacy and disease progression in patients with metastatic angiosarcoma.

We thank the reviewer for the insightful comments and positive feedback. We have added more detailed information about the 25 genes in Supplementary Table 2B. We have also expanded the discussion about the relevance of sequencing the recurrence.

Reviewer 2 Report

Comments and Suggestions for Authors

In this case study, authors present a long-term breast cancer patient who lately developed angiosarcoma. Patient went through two surgeries: first for the resection of angiosarcoma and the second for arm amputation. Patients then received 6 cycles of chemotherapy. Interestingly she showed complete response after the 5th cycle and treatment was paused after the 6th cycle. Unfortunately, patient then developed metastasis in the lung and vertebrae which was shown to be angiosarcoma. Patient then received three weeks of radio therapy before she dies.

Blood draws before and after the second surgery together with those from different treatment cycles were collected. Whole genome sequencing for angiosarcoma primary tumor detected 25 disease specific mutations. The frequency of these mutations was then detected in cfDNA from all blood draws. Interestingly, number of cfDNA increased after second surgery and kept high during all time of treatment including the time which radiological investigations indicated complete response.   

This case is interesting and supports previous studies which found that numbers of circulating tumor cells (CTCs) are high at time when CT scans show tumor remission. Unfortunately, later these tumors showed swift progression. These findings suggest that traditional radiological investigations should be complemented by liquid biopsy.  The case is very well presented and the fact that data could distinguish between the two possible sources of cfDNA (Angiosarcoma Vs. Breast Cancer) using disease specific mutations is impressive. However, I have one comment.

The observed increase in cfDNA on the day after surgery could be due to the surgery itself, where additional cfDNA moclecules could be introduced by damaged cells during surgery. It has been previously shown that CTCs numbers increase after surgery and could stay circulating for over a week before it clears off. Could author provide a comparison between cfDNA quantity from a week or two after surgery and that from just before the initiation of therapy?

Similarly, the increased cfDNA during chemo and radio therapy (including the remission time) could be due to the ongoing apoptosis because of therapy. If this is the case, it means that quantity of cfDNA could not be a decisive indicator for therapy response as it could be an indicator for a good response (if the source of cfDNA is cell death). On the other side, it could be an indicator for a poor response (if the source of cfDNA is unresponsive tumor or un detected metastatic lesions. Could authors discuss, please?

Author Response

Comments and Suggestions for Authors

In this case study, authors present a long-term breast cancer patient who lately developed angiosarcoma. Patient went through two surgeries: first for the resection of angiosarcoma and the second for arm amputation. Patients then received 6 cycles of chemotherapy. Interestingly she showed complete response after the 5th cycle and treatment was paused after the 6th cycle. Unfortunately, patient then developed metastasis in the lung and vertebrae which was shown to be angiosarcoma. Patient then received three weeks of radio therapy before she dies.

Blood draws before and after the second surgery together with those from different treatment cycles were collected. Whole genome sequencing for angiosarcoma primary tumor detected 25 disease specific mutations. The frequency of these mutations was then detected in cfDNA from all blood draws. Interestingly, number of cfDNA increased after second surgery and kept high during all time of treatment including the time which radiological investigations indicated complete response.   

This case is interesting and supports previous studies which found that numbers of circulating tumor cells (CTCs) are high at time when CT scans show tumor remission. Unfortunately, later these tumors showed swift progression. These findings suggest that traditional radiological investigations should be complemented by liquid biopsy.  The case is very well presented and the fact that data could distinguish between the two possible sources of cfDNA (Angiosarcoma Vs. Breast Cancer) using disease specific mutations is impressive. However, I have one comment.

We thank the reviewer for the positive feedback and useful comments.

The observed increase in cfDNA on the day after surgery could be due to the surgery itself, where additional cfDNA moclecules could be introduced by damaged cells during surgery. It has been previously shown that CTCs numbers increase after surgery and could stay circulating for over a week before it clears off. Could author provide a comparison between cfDNA quantity from a week or two after surgery and that from just before the initiation of therapy?

Indeed, this is an insightful comment. We fully agree with the reviewer. Unfortunately, we have no additional blood samples during this time period. However, we have updated Figure 2 with subplot 2D showing the level of cell-free DNA analyzed after plasma extraction. We see an increase in total cfDNA level after surgery mainly due to cell death of non-tumor cells. We have updated Results and Discussions with this additional information (Lines 172-175 and lines 240-246).

Similarly, the increased cfDNA during chemo and radio therapy (including the remission time) could be due to the ongoing apoptosis because of therapy. If this is the case, it means that quantity of cfDNA could not be a decisive indicator for therapy response as it could be an indicator for a good response (if the source of cfDNA is cell death). On the other side, it could be an indicator for a poor response (if the source of cfDNA is unresponsive tumor or un detected metastatic lesions. Could authors discuss, please?

This is an interesting suggestion and complex issue, not only for this study but for the ctDNA field in general. We cannot know for sure. Normally, in patients without treatment with high tumor cell proliferation the level of ctDNA is high, while tumors with minimal proliferating cells display low or no ctDNA despite a tumor burden (based on other tumor entities). Normally, the ctDNA level decreases rapidly to low or non-detectable levels during chemotherapy. If not, the patient is a poor-responder (our own unpublished data, study in progress). However, in angiosarcoma and this case we cannot say for sure. We have added a sentence of caution in the Discussion (Lines 250-251) that we cannot determine if the detected ctDNA is due to tumor progression or the treatment. With the new data in Figure 2D, we can also determine that total cfDNA level increases after radiotherapy and that a large part of it is due to increased ctDNA release. We have updated the manuscript with this new information (Lines 172-175 and lines 246-248)